Subject Area:
molecular biology

Keywords:
model organisms, transcriptional regulation, *Caenorhabditis elegans*, *Drosophila melanogaster*

Authors for correspondence:
Xiao Zhu
e-mail: bioxzhu@yahoo.com
Hui Luo
e-mail: luohui@gdmu.edu.cn

†These authors contributed equally to the study.

# Transcriptional regulation in model organisms: recent progress and clinical implications

Jiaqi Tang[1,2,†], Zhenhua Xu[3,†], Lianfang Huang[1,2], Hui Luo[1] and Xiao Zhu[1,2]

[1]The Marine Biomedical Research Institute, Southern Marine Science and Engineering Guangdong Laboratory Zhanjiang, Guangdong Medical University, Zhanjiang 524023, People's Republic of China
[2]Guangdong Key Laboratory for Research and Development of Natural Drugs, Zhanjiang 524023, People's Republic of China
[3]Center for Cancer and Immunology, Brain Tumor Institute, Children's National Health System, Washington, DC 20010, USA

XZ, 0000-0002-1737-3386

In this review, we will summarize model organisms used by scientists in the laboratory, including *Escherichia coli*, yeast, *Arabidopsis thaliana*, nematodes, *Drosophila*, zebrafish, mice and other animals. We focus on the progress in research exploring different types of *E. coli* in the human body, and the specific molecular mechanisms by which they play a role in humans. First, we discuss the specific transcriptional regulation mechanism of *E. coli* in cell development, maturation, ageing and longevity, as well as tumorigenesis and development. Then, we discuss how the synthesis of some important substances in cells is regulated and how this affects biological behaviour. Understanding and applying these mechanisms, presumably, can greatly improve the quality of people's lives as well as increase their lifespan. For example, some *E. coli* can activate certain cells by secreting insulin-like growth factor-1, thus activating the inflammatory response of the body, while other *E. coli* can inactivate the immune response of the body by secreting toxic factors.

## 1. Introduction

It is well known that life is orchestrated by many signals in the body that are transcribed by our genes. Thereafter, transcriptional regulation is of great significance in maintaining normal intracellular activities and homeostasis. Model organisms are those species that have been extensively studied and for which researchers have a deep understanding of their biological phenomena. Based on the results obtained in the studies of these model organisms, some models covering many organisms can often be summarized and applied in various fields of research. In the laboratory, model organisms are the most favoured research objects for researchers because they are easy to obtain, are easy to raise and have short reproduction cycles [1]. Because of these precise characteristics, model organisms are favoured by laboratories with limited funds. At the same time, the model biology experiment can also shorten the experimental cycle, so that scientists can find the answer to a question in the shortest time, such as the molecular mechanism of the occurrence of a disease, the molecular mechanism of drug resistance and so on [2]. Investigating these molecular mechanisms could help in the development of new treatments for these diseases, and may even make some incurable diseases 'treatable'. For example, by exploring the mechanism by which some of the mutant strains convert certain substances into specific products, we can create engineered bacteria that can treat waste and even turn waste into useful products, thereby recycling the material [3,4]. And by looking at some of the molecular mechanisms that affect longevity, we might be able to extend the lifespan. Thus it can be seen that these pathways have important effects on our survival and life, and

royalsocietypublishing.org/journal/rsob    Open Biol. 9: 190183

exploration of these mechanisms is necessary so that we can take advantage of them (figure 1 and table 1). In this review, we will discuss progress in the research on the regulation of transcription in model organisms.

## 2. Escherichia coli

Previous studies have found that human infection can cause skeletal muscle loss and fat loss [31,32]. Recently, scientists have discovered that *E. coli* promotes human secretion of insu-lin-like growth factor-1 (IGF-1), and then repairs damaged tissue [5].

But some studies have found that *E. coli* T3SS (type III secretion system) can sense the host and respond by remodelling gene expression to secrete virulence factors to antagonize the host cell immune response [6–9]. These two studies have discovered the mechanism affecting cell repair and division and that inhibiting immune response, which is of great significance.

Recently, scientists have found that overexpression of *MetR* and *MdlB* genes can protect *E. coli* from prenyl alcohol. *MetR* can express biosynthesis regulators, while *MdlB* gene expression products are transporters. Both products can increase the yield of methyl-2-buten-1-ol [33]. Understanding transcriptional regulation in *E. coli* has enabled the use of microbial host platforms for the production of bulk commodities (figure 2 and table 1).

## 3. Yeast

Recently, a research group found that *cleavage and polyadeny-lation factor* (CPF) could add a poly(adenylate) tail to yeast mRNA [10]. Genes generally contain multiple polyadenylate sites (PASs). By using different PASs, cells produce 3′ untranslated regions (3′UTR) of different lengths and isoforms with different coding sequences, a process known as variable addendum [34,35].

A group of researchers demonstrated that the 3′ proces-sing complex *CFI* (*cleavage factor I*) can regulate variable tails by associating with RNA polymerase II (RNAPII) core promoter regions. The researchers found that *CFI* consists mainly of two proteins, *CFI-25* and *CFI-68*, and found that these two proteins have the function of enhancing the remote 3′-end polyadenylation site [36–38]. Studies have found that mRNA with a long 3′ non-translation region is generally concentrated in the nucleus [39,40]. However, the short 3′UTR subtype had more gene expression when upre-gulated, while the long 3′UTR subtype had the opposite. This is because the selection of poly sites depends on the tran-scriptional activity of the gene promoter. Other researchers have found that RNAPII elongation rates slow down, leading to preferential use of proximal PASs. It can be seen that pro-moter activity and the transcriptional elongation rate can also influence the variable tail addition process [41–43]. Currently, some research groups have found that *ALYREF* can bind to the 3′ end of mRNA [11]. Recently, a research group has suc-cessfully revealed the mechanism. The researchers found that inhibition of mRNA export of nuclear receptor *NXF1* or mRNA export complex TREX components and ALYREF expression will lead to the proximal PASs being used prefer-entially, resulting in shorter mRNA isomers in the 3′UTR. The researchers also found two factors that could influence the variable tail addition—the length of the gene and the

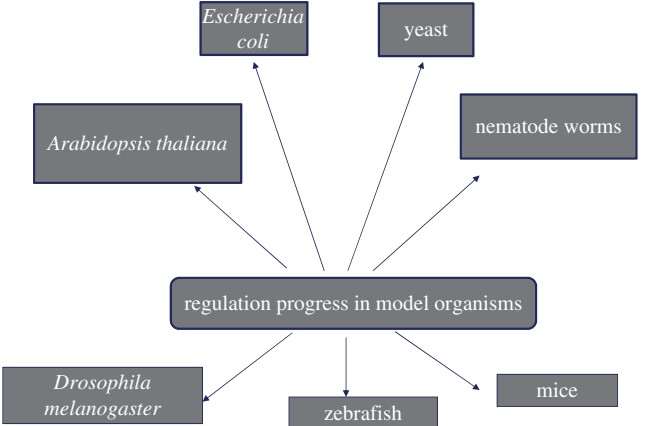

**Figure 1.** The common model organisms in this review.

abundance of AT in the 3′ exon [11,44–46]. In addition, the researchers also found that NXF1 can directly interact with 3′ processing factor CFI-6 [38,47]. NXF1 typically binds to active transcribed genes and then interacts with RNAPII to complete its extension at the 3′ end of the gene. These researchers have made an in-depth study of the process of variable tail addition, discovering not only many factors that can affect variable tail addition but also the mechanism that determines the allosteric property.

Recently, a team found that msn2/4 protein produced by mitochondria can control the whole cell by using the intracellu-lar organelle protein transcription programme which it relies on. When this programme breaks down, it results in a loss of cellular function [12]. This study revealed the regulatory role of transcription factors in cell function, highlighting the important role that determines regulation in cell function (table 1).

## 4. Arabidopsis thaliana

Scientists recently found that Polycomb in *Arabidopsis* can induce gene silencing [13]. Polycomb protein is mainly com-posed of PRC1 and PRC2, in which PRC2 binds to specific transcription factors and locates the DNA regions where these transcription factors bind. This process uses Polycomb response elements (PREs). Subfamilies of transcription factors that bind to PREs can co-localize with PRC2 on chromatin, thus inducing gene silencing [14]. While studying gene sequences regulated by PRC2, researchers found that six motifs bind to different types of transcription factors, some of which were also associ-ated with the distribution of Polycomb in *Arabidopsis* (table 1). In *Arabidopsis*, PRC2 protein complexes were recruited to DNA sequences containing PREs by binding *BPC* (*basic penta cysteine*) and *ZnF* (*zinc finger*) subfamily transcription factors, promoting histone methylation in nearby regions and inhibiting transcription of downstream proteins [15,17,18,48,49]. Thus it can be seen Polycomb could affect the expression of plant proteins, and it even may increase the protein content in fruits.

Nitrogen is an essential element in plant growth and metabolism. Recently, studies have found that nitrogen trans-porters, assimilation enzymes and signal transduction factors are regulated by transcription and can respond to changes in nitrogen availability [50,51]. Previously, 16 tran-scription factors were found to play a role in nitrogen metabolism in *A. thaliana* [52,53]. A study found that the nitrogen metabolism of plant transcription regulatory

**Table 1.** The summary functions of model organisms in transcriptional regulation.

| kingdom | model organisms | functions |
|---|---|---|
| Prokaryotes | E. coli | E. coli cells secrete transcriptional products that control the process of growth, repair and maturation. E. coli can also inhibit the growth of some harmful microorganisms or hinder the damaging effects of harmful microbes in humans, so as to maintain the stability of the internal environment [5–9]. |
| | Saccharomyces cerevisiae | Yeast is favoured by scientists because of its simple structure. It is often used by scientists to study the structure and function of genes and the synthesis of RNA. For example, the effects of CPF and ALYREF on RNA synthesis were investigated. In addition, yeast is often used to study how organelles affect cell status. For example, to explore how mitochondria activate protective programmes when they are under stress to ensure that all cellular functions are under control [10–12]. |
| Plants | A. thaliana | We know that nitrogen is an essential nutrient element in plant growth and basic metabolism. And Arabidopsis can synthesize a variety of transcription factors to regulate their own nitrogen metabolism, and the effects of transcription factors related to nitrogen metabolism enable individual yeast hybrid network (yeast hybrid network for nitrogen—associated with metabolism, YNM) genes to influence plant hormone secretion, and regulate its growth and metabolic processes. In addition, the YNM network can also predict the effects of transcription factors on shoot growth, and thus control the growth and maturation time of plants. Polycomb protein in Arabidopsis can inhibit gene expression, and PRC2 protein complex in Polycomb can also promote histone methylation (H3K27me3) in nearby regions, thus inhibiting the transcription of downstream proteins [13–18]. |
| Animals | Caenorhabditis elegans | During a study on nematode worms, researchers found that transcription expression products, including Krupp transcription factor (KLF), bcat-1 enzyme, H3K4me3 methyl transferase, etc., can affect the lifespan of nematode worms through various processes. For example, the KLF protein can block the ageing process of cells by regulating autophagy, and it can prevent age-related loss of blood vessel function, thus preventing hypertension, Alzheimer's disease and other diseases [19–21]. |
| | Drosophila melanogaster | In a fruit fly study, researchers found that many transcriptional expression products can affect the behaviour of fruit flies. For example, melanochemical-induced neuropeptides can control the heartbeat of fruit flies and affect the ejaculation ability of male fruit flies. Xylose isomerase can reduce the activity of neurons that produce the neurotransmitter octopamine, which inhibits the movement of fruit flies [22–24]. |
| | zebrafish | Zebrafish as experimental models, in addition to being used for screening of drug targets, such as screening natural products with anti-angiogenesis potential, are often used to study some molecular mechanisms; for example, in a zebrafish model, researchers found that the thymus cells Foxn1 by MCM2 maintain thymic epithelial cells and T-cell development and help to control the opening of important genes [25–27]. |
| | mouse | Mice are widely used in experiments because they are mammals like humans and many mechanisms of the human body are also present in mice. For example, the role of transcription factors Hobit and Blimp1 in memory T cells was explored to investigate the effect of NFIL3 transcription factor expression on fat metabolism and the anti-inflammatory mechanism that can naturally repair the eyes [4,28–30]. |

networks regulates the architecture of root and shoot systems in response to changes in nitrogen availability. In the network, the enzymes involved in nitrogen metabolism are transcriptionally modified by feedback via genetic perturbation of the nitrogen metabolism. Nitrogen-regulated transcription factors bind to promoters of hormone-regulating genes whose expression affects rosette size and flowering time [54]. Auxin has also been shown to promote cell elongation, but at higher concentrations it does the opposite [55]. Recently, a research group found that auxin can be sensed by the auxin receptor complex on the cell membrane [56]. When higher concentrations of auxin bind to Arabidopsis kinase TMK1 receptors on the curved surface of the plasma membrane, it activates signalling pathways that inhibit cell growth, which in turn inhibits cell elongation [57]. These studies enable us to understand the mechanisms involved in plant cell growth, which are very helpful in increasing agricultural productivity.

Recently, some research groups found that Arabidopsis contains drought-inducible MzSNAT5, an isoform of the

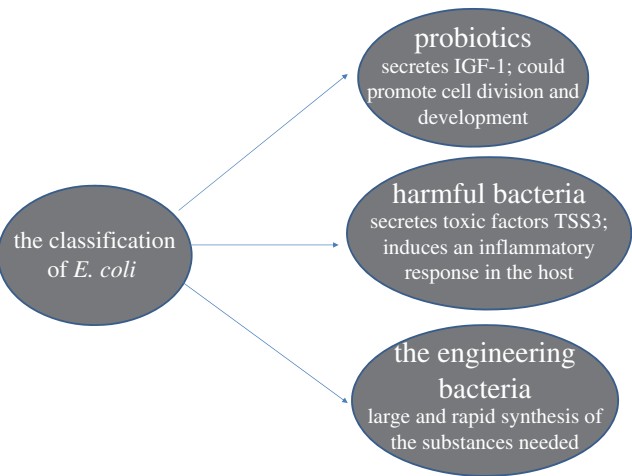

**Figure 2.** Three types of *E. coli* which play different roles.

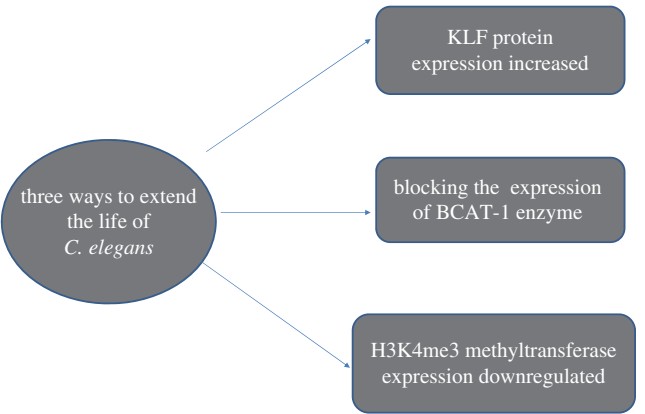

**Figure 3.** Three pathways that extend the lifespan of *C. elegans*. By upregulating the expression of Krupp transcription factor and downregulating the expression of BCAT-1 enzyme and H3K4me3 methyl transferase, the lifespan of the nematode could be extended.

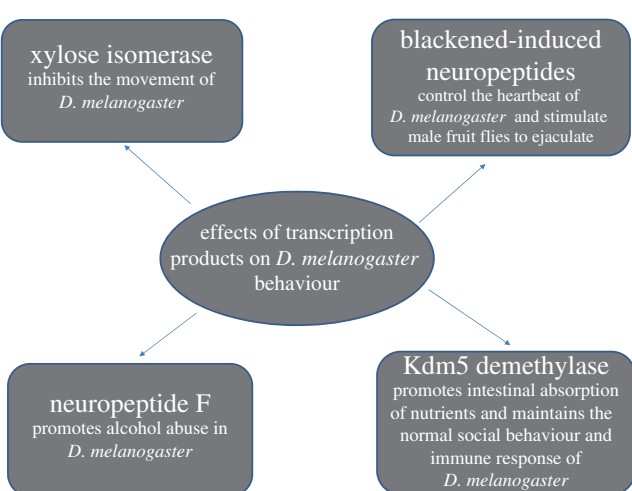

**Figure 4.** Effects of transcriptional products on ejaculation, alcohol abuse, exercise and other behaviours of *Drosophila melanogaster*.

sodium ion-dependent neutral amino acid transporter (SNAT), and its isotopic expression promotes the synthesis and secretion of melatonin, thereby enhancing drought tolerance [58]. Another team found that drought-induced long non-coding RNAs (DRIR) molecules enhance *Arabidopsis* resistance to drought and salt stress [59]. These two studies have found different mechanisms that can enhance drought resistance of *A. thaliana*, which has important reference significance for enhancing drought resistance of other crops.

## 5. Caenorhabditis elegans

Recently, scientists discovered that, when Krupp transcription factor (*KLF*) is overexpressed in nematodes, the worms have a longer lifespan. *KLF4* overexpression was found to delay vessel ageing and to enhance autophagy in aged mice. But *KLF* levels decrease with age in human vascular endothelium [19]. This suggests that keeping *KLF* levels constant may extend the lifespan of nematodes or humans (table 1).

Another team found that blocking the expression of the *branched-chain amino acid transferase-1* (*BCAT-1*) gene caused excessive levels of branched-chain amino acids (BCAAs). The increased levels of BCAAs promote *C. elegans* longevity by reducing a LET-363/mTOR-dependent neuro-endocrine signal, DAF-7/TGFb DAF-7/TGFb, in a cell-non-autonomous manner [20]. Another research group found that a deficiency in H3K4me3 methyltransferase altered fat metabolism with a specific enrichment of mono-unsaturated fatty acids in the intestine and extends the lifespan of worms [21]. These findings are very helpful in maintaining health and extending life span (figure 3).

## 6. Drosophila melanogaster

Recently, researchers have found that chromosomal instability (CIN)-induced invasive cells activate the ERK and JNK signalling pathways through the actin cytoskeleton, triggering an invasive transcriptional programme that is performed by the oncogene Fos and inhibited by the tumour suppressor Capicua [60]. This finding points to a new direction to stop the metastasis of cancer cells [61].

Recently, a research group found that, after being infected by bacteria, fruit flies overexpress NEMURI, an anti-microbial peptide that can be secreted ectopically to drive prolonged sleep, thereby improving the body's resistance and survival rate and increasing the length and depth of sleep [62,63]. This suggests that there is a link between sleep and immunity, which might be determined by the length and depth of sleep.

Recently, scientists found that xylose isomerase from gut bacteria lowers trehalose levels in fruit flies, which in turn reduces the neuronal activity of the synthesis of octopamine and ultimately inhibits motor behaviour in fruit flies [22]. Other research groups have found that melanosis-inducing neuropeptides expressed in the abdomen of male fruit flies control their heartbeat and ejaculation, and activating neuropeptide corazonin (CRZ)/Crz-receptor signalling increased neuropeptide F (npf) transcript levels, serving as an essential part of the mating reward mechanism in *Drosophila* [23,64]. A team found that male fruit flies rejected by female fruit flies abused alcohol, which is caused by the reduced NPF levels secreted by the brain [24]. Another team found that flies deficient in kdm5 exhibited gut dysbiosis, abnormal social behaviour and abnormal immune activation [65]. These studies revealed the effects of transcriptional products on various behaviours of *D. melanogaster*, reflecting the importance of transcriptional regulation in maintaining normal behaviours (figure 4 and table 1).

# 7. Zebrafish

At present, scientists have successfully established the transplanting tumour zebrafish model [66] and zebrafish non-infectious intestinal inflammation model [67]. These zebrafish models have many advantages [68] and can be used to explore tumour growth, metastasis mechanisms and mechanisms of disease development, as well as screening anti-tumour drugs and evaluating new drugs [69]. Recently, a research team used the zebrafish model to evaluate the anti-angiogenic drug mundoserone. The researchers found that mundoserone could inhibit zebrafish blood vessel formation by downregulating SLIT3/ROBO1 and FGFR/PTP-RB, as well as upregulating NOTCH1A [25].

A few years ago, scientists found in zebrafish models that Foxn1 from thymic cells regulated interaction of thymic epithelial cells (TECs) and promoted T-cell development via MCM2 (table 1) [26]. Another group created a transgenic model for conditional inducible Foxn1 expression in TECs and found that forced upregulation of FOXN1 can substantially reverse age-related thymic involution and improve thymus function [27]. Thus, Foxn1 plays a very important role in the development of the immune system and organs. Although these models are widely used, they still have significant limitations [70]. Therefore, these models still need to be further refined.

# 8. Mice

Recently, a research group found that the bacteria in the intestinal tract of mice can secrete certain substances to act on intestinal immune cells, specifically inhibiting the expression of rev-erba protein, thereby improving the transcription of NFIL3, and finally promoting the uptake and storage of fat by intestinal epithelial cells [28]. Another team found that dieting altered the composition of the mice's gut flora and even extended their lives. Dieting suppressed expression of the key bacterial enzymes necessary for lipid A biosynthesis, a critical lipopolysaccharide (LPS) building component. As a result, the amount of LPS in the body was reduced. Genetic and pharmacological block of the binding of LPS to its receptor Toll-like receptor 4 promoted adipose tissue browning and decreased fat accumulation, associated with improved glucose tolerance and insulin sensitivity [71]. These research results have very important implications for the development of new methods for weight loss.

Recently scientists found that mesencephalic astrocyte-derived neurotrophic factor (MANF) could promote the conversion of 'pro-inflammatory immune cells' into repair immune cells in mice, improving retinal repair capacity [72]. Injecting MANF may help to restore vision to those people whose vision is impaired by retinal damage.

In recent years, the incidence of tumours has been continuously increasing [73]. We know that both inbred mice and immunodeficient mice are mature models of transplanted tumours [74]. Recently, scientists found that the Hobit and Blimp1 genes in mouse models can control a transcription process that promotes immune cells to move to cancer and infected tissues. The scientists also found increased levels of the transcription factor Hobit in resident memory T cells; together with Blimp1, Hobit regulates the development of tissue-resident memory T cells (table 1)

[29]. TECs are required both for thymus organogenesis and for promotion of thymocyte maturation. Another study group found that mature T cells interact with TECs, and this interaction is required for the late stage of TEC differentiation [75]. Another team also found that knocking out two genes, Bmal1 and Per2, caused tumours to grow faster. This is because the two genes inhibit the expression of c-Myc protein, which could enhance cell metabolism and proliferation. Therefore, when the expression of Bmal1 and Per2 genes is downregulated, tumour cells will grow and proliferate rapidly [30]. These results are of great significance for the treatment of immunodeficiency and the control of tumour evolution.

# 9. Discussion and perspectives

From the above studies, we can see that differences in the gut microbiota composition can affect energy metabolism. Also *E. coli* may be beneficial to the human body and can also be harmful. This depends mainly on the role of *E. coli* and its transcription products in the human body. We can also see that transcriptional expression products of cells and organelles can affect the growth, development, maturation and senescence of an organism. For example, Msn2/4 protein produced by mitochondria can affect ageing of the body; Kepulu transcription factor can maintain the autophagy ability of cells. The Beat-1 enzyme and H3K4me3 enzyme can affect certain metabolic processes in the body. In addition, signalling pathways can also regulate the physical activity of certain model organisms, such as ERK and JNK signalling pathways, which can induce invasive behaviour in *Drosophila* cells, while melanogenesis-induced neuropeptides can affect the frequency of heartbeats in *Drosophila* and stimulate male fruit fly ejaculation. There are some transcription factors that can even affect the function of the immune system. For example, the transcription factors Blimp and Hobit can affect the development of T cells, which in turn affects the body's ability to resist infection. Thus, transcriptional regulation plays a very important role in life activities.

If the above research results can be applied to clinical practice, it is believed that it would be very helpful for the treatment of disease. For example, by upregulating the expression of transcription factors Blimp and Hobit, T-cell development will be accelerated, so an organism may be able to resist invasion by a tumour. Melatonin-induced neuropeptides can affect human heart rate, so we can use neuropeptides to control the heart rate. Thus, these transcription factors are 'switches' in various reactions in the body. By controlling these switches, we can control the extent and speed of metabolic processes that affect cell growth, differentiation, maturation and death. This may be used as treatment for serious diseases, and we believe that with the deepening of research, sooner or later, humans will be able to overcome cancer and other incurable diseases. It can be seen that the normal operation of transcriptional regulation is a necessary condition for the organism to survive healthily.

Data accessibility. This article has no additional data.
Authors' contributions. J.T., Z.X., L.H. and X.Z. edited selected paragraphs, J.T. and Z.X. participated in the preparation of the table and figures, X.Z. and H.L. designed and edited the whole text.
Competing interests. We have no competing interests.

Funding. This work was supported by the Southern Marine Science and Engineering Guangdong Laboratory Zhanjiang (ZJW-2019-07). This work was also supported partly by National Natural Science Foundation of China (81541153); Guangdong Provincial Science and Technology Department (2016A050503046, 2015A050502048 and 2016B030309002); and The Public Service Platform of South China Sea for R&D Marine Biomedicine Resources (GDMUK201808). The funders had no role in the study design, data collection and analysis, or in the preparation and publication of the manuscript.

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
