## [Reviewer comments · Open Biology]

Review History

RSOB-19-0183.R0 (Original submission)

Review form: Reviewer 1

Recommendation

Accept with minor revision (please list in comments)

Do you have any ethical concerns with this paper?

No

Comments to the Author

The review article entitled 'Transcriptional regulation in model organisms: recent progress and clinical implications' is mainly discussed from several aspects, including *E. coli*, yeast, *Arabidopsis thaliana*, nematodes, *Drosophila*, zebrafish, mice and other animals. This is a relatively well-written review. The references cited are thorough and most updated results are included.

Some concern/suggestions, listed as below, may strengthen this review.

1. Table 1, references are needed.
2. Spelling, Insulin-like growth factor-1 (IGF-1)
3. "...keeping cells from dividing and repairing damaged tissue, leading to muscle and fat loss.", it is better to add a reference here.
4. But some studies...[6], better to cite more than one reference.
5. "Both products can increase the yield of Prenyl alcohol [7]." Please double check ref. 7 to ensure it is Prenyl alcohol.
6. Rewrite these sentence "... cleavage and polyadenylation factor (CPF) could add poly (adenylate) tail to yeast mRNA." "Can achieve do not exercise can lose weight," "we can gut microbiota composition can affect energy"...
7. "Discussion and prospectives" section need to strengthen author's point of view.
8. Expressions of some sentences are colloquial. In some paces, the manuscript needs editing for English usage error and statement accuracy

Decision letter (RSOB-19-0183.R0)

28-Oct-2019

Dear Dr Zhu

We are pleased to inform you that your manuscript RSOB-19-0183 entitled "Transcriptional regulation in model organisms: recent progress and clinical implications" has been accepted by the Editor for publication in Open Biology. The reviewer(s) have recommended publication, but also suggest some minor revisions to your manuscript. Therefore, we invite you to respond to the reviewer(s)' comments and revise your manuscript.

Please submit the revised version of your manuscript within 14 days. If you do not think you will be able to meet this date please let us know immediately and we can extend this deadline for you.

- 1) A text file of the manuscript (doc, txt, rtf or tex), including the references, tables (including

captions) and figure captions. Please remove any tracked changes from the text before submission. PDF files are not an accepted format for the "Main Document".

2) A separate electronic file of each figure (tiff, EPS or print-quality PDF preferred). The format should be produced directly from original creation package, or original software format. Please note that PowerPoint files are not accepted.

3) Electronic supplementary material: this should be contained in a separate file from the main text and meet our ESM criteria (see <http://royalsocietypublishing.org/instructions-authors#question5>). All supplementary materials accompanying an accepted article will be treated as in their final form. They will be published alongside the paper on the journal website and posted on the online figshare repository. Files on figshare will be made available approximately one week before the accompanying article so that the supplementary material can be attributed a unique DOI.

Online supplementary material will also carry the title and description provided during submission, so please ensure these are accurate and informative. Note that the Royal Society will not edit or typeset supplementary material and it will be hosted as provided. Please ensure that the supplementary material includes the paper details (authors, title, journal name, article DOI). Your article DOI will be 10.1098/rsob.2016[last 4 digits of e.g. 10.1098/rsob.20160049].

4) A media summary: a short non-technical summary (up to 100 words) of the key findings/importance of your manuscript. Please try to write in simple English, avoid jargon, explain the importance of the topic, outline the main implications and describe why this topic is newsworthy.

Images

Data-Sharing

It is a condition of publication that data supporting your paper are made available. Data should be made available either in the electronic supplementary material or through an appropriate repository. Details of how to access data should be included in your paper. Please see <http://royalsocietypublishing.org/site/authors/policy.xhtml#question6> for more details.

Data accessibility section

Sincerely,

The Open Biology Team
<mailto:openbiology@royalsociety.org>

Reviewer(s)' Comments to Author:

Referee: 1

Comments to the Author(s)

The review article entitled 'Transcriptional regulation in model organisms: recent progress and clinical implications' is mainly discussed from several aspects, including *E. coli*, yeast, *Arabidopsis thaliana*, nematodes, *Drosophila*, zebrafish, mice and other animals. This is a relatively well-written review. The references cited are thorough and most updated results are included.

Some concern/suggestions, listed as below, may strengthen this review.

1. Table 1, references are needed.
2. Spelling, Insulin-like growth factor-1 (IGF-1)
3. "...keeping cells from dividing and repairing damaged tissue, leading to muscle and fat loss.", it is better to add a reference here.
4. But some studies...[6], better to cite more than one reference.
5. "Both products can increase the yield of Prenyl alcohol [7]." Please double check ref. 7 to ensure it is Prenyl alcohol.
6. Rewrite these sentence "... cleavage and polyadenylation factor (CPF) could add poly (adenylate) tail to yeast mRNA." "Can achieve do not exercise can lose weight," "we can gut microbiota composition can affect energy" ...
7. "Discussion and prospectives" section need to strengthen author's point of view.
8. Expressions of some sentences are colloquial. In some paces, the manuscript needs editing for English usage error and statement accuracy

Author's Response to Decision Letter for (RSOB-19-0183.R0)

See Appendix A.

Decision letter (RSOB-19-0183.R1)

31-Oct-2019

Dear Dr Zhu

We are pleased to inform you that your manuscript entitled "Transcriptional regulation in model organisms: recent progress and clinical implications" has been accepted by the Editor for publication in *Open Biology*.

Article processing charge

Please note that the article processing charge is immediately payable. A separate email will be sent out shortly to confirm the charge due. The preferred payment method is by credit card; however, other payment options are available.

Sincerely,

The Open Biology Team
mailto: openbiology@royalsociety.org

Appendix A

Response to the The Reviewer 1

Thank you for your valuable comments. We correct all you mentioned in the text and the Table

1. The references were added in “Table 1”.
2. We corrected the word IGF-1 in the “Abstract”.
3. “Refs 5 and 6” were added according your suggestion.
4. “Refs 9-11” were added according your suggestion.
5. The word “Prenyl alcohol” was changed to its official name “methyl-2-buten-1-ol”.
6. We have re-written the three sentences. Please see “Line 2 Page 5“, “Line 10 Page 14” and “Line 13-14 Page 14”.
7. We strengthen the “Discussion and prospectives” section, see “The first and the last sentences”.
8. According to the Reviewer’s suggestion, in order to express more accurately, we corrected some grammar error and several sentences. For example “Line 7 – 8 Page 5”.